# The Effectiveness of Cobalamin (B12) Treatment for Autism Spectrum Disorder: A Systematic Review and Meta-Analysis

**DOI:** 10.3390/jpm11080784

**Published:** 2021-08-11

**Authors:** Daniel A. Rossignol, Richard E. Frye

**Affiliations:** 1Rossignol Medical Center, 24541 Pacific Park Drive, Suite 210, Aliso Viejo, CA 92656, USA; 2Barrow Neurological Institute at Phoenix Children’s Hospital, 1919 E Thomas Rd, Phoenix, AZ 85016, USA; rfrye@phoenixchildrens.com

**Keywords:** autism spectrum disorder, cobalamin, glutathione, methylation, methylcobalamin, redox metabolism

## Abstract

Autism spectrum disorder (ASD) is a common neurodevelopmental disorder affecting 2% of children in the United States. Biochemical abnormalities associated with ASD include impaired methylation and sulphation capacities along with low glutathione (GSH) redox capacity. Potential treatments for these abnormalities include cobalamin (B12). This systematic review collates the studies using B12 as a treatment in ASD. A total of 17 studies were identified; 4 were double-blind, placebo-controlled studies (2 examined B12 injections alone and 2 used B12 in an oral multivitamin); 1 was a prospective controlled study; 6 were prospective, uncontrolled studies, and 6 were retrospective (case series and reports). Most studies (83%) used oral or injected methylcobalamin (mB12), while the remaining studies did not specify the type of B12 used. Studies using subcutaneous mB12 injections (including 2 placebo-controlled studies) used a 64.5–75 µg/kg/dose. One study reported anemia in 2 ASD children with injected cyanocobalamin that resolved with switching to injected mB12. Two studies reported improvements in markers of mitochondrial metabolism. A meta-analysis of methylation metabolites demonstrated decreased S-adenosylhomocysteine (SAH), and increased methionine, S-adenosylmethionine (SAM), SAM/SAH ratio, and homocysteine (with small effect sizes) with mB12. Meta-analysis of the transsulfuration and redox metabolism metabolites demonstrated significant improvements with mB12 in oxidized glutathione (GSSG), cysteine, total glutathione (GSH), and total GSH/GSSG redox ratio with medium to large effect sizes. Improvements in methylation capacity and GSH redox ratio were significantly associated with clinical improvements (with a mean moderate effect size of 0.59) in core and associated ASD symptoms, including expressive communication, personal and domestic daily living skills, and interpersonal, play-leisure, and coping social skills, suggesting these biomarkers may predict response to B12. Other clinical improvements observed with B12 included sleep, gastrointestinal symptoms, hyperactivity, tantrums, nonverbal intellectual quotient, vision, eye contact, echolalia, stereotypy, anemia, and nocturnal enuresis. Adverse events identified by meta-analysis included hyperactivity (11.9%), irritability (3.4%), trouble sleeping (7.6%), aggression (1.8%), and worsening behaviors (7.7%) but were generally few, mild, not serious, and not significantly different compared to placebo. In one study, 78% of parents desired to continue mB12 injections after the study conclusion. Preliminary clinical evidence suggests that B12, particularly subcutaneously injected mB12, improves metabolic abnormalities in ASD along with clinical symptoms. Further large multicenter placebo-controlled studies are needed to confirm these data. B12 is a promising treatment for ASD.

## 1. Introduction

Autism spectrum disorder (ASD) is a common neurodevelopmental disorder affecting 2% of children in the United States [1]. ASD is defined behaviorally by reduced social communication and the existence of restrictive and repetitive behaviors and interests. Although ASD is currently classified as a psychiatric disorder, a number of medical comorbidities and biochemical abnormalities are associated with ASD and may contribute to symptoms. These include immune disorders [2], mitochondrial dysfunction [3,4], oxidative stress [5], seizures [6], gastrointestinal problems [7], and impaired methylation capacity [8].

The methylation cycle recycles homocysteine to methionine, the precursor of S-adenosylmethionine (SAM), which is the major methyl donor for many chemical processes in the body including DNA methylation. Importantly, the methylation cycle is intricately connected to redox metabolism through the contribution of homocysteine (Figure 1). Homocysteine is metabolized to cystathionine and then to cysteine, which is the rate-limiting precursor of glutathione (GSH), the major antioxidant in the body.

The first study to formally evaluate methylation and redox capacity in ASD reported significantly lower plasma levels of methionine, SAM, homocysteine, cystathionine, cysteine, and total GSH, and significantly higher concentrations of S-adenosylhomocysteine (SAH), adenosine, and oxidized GSH (GSSG) in 20 children with ASD compared to 33 controls. The lower SAM/SAH ratio is indicative of impaired methylation capacity and a lower GSH with elevated GSSG represents impaired redox capacity [8]. These initial findings have been confirmed in several other prospective, controlled studies in blood [9,10,11,12] and brain samples [13,14,15], and a meta-analysis has confirmed the consistency of these findings across multiple studies and multiple laboratories [5]. A meta-analysis of 22 studies confirmed a consistent finding of impaired DNA methylation in ASD [16] and impaired DNA methylation has been associated with mitochondrial dysfunction in ASD [17]. These biochemical abnormalities have even been proposed to be diagnostic of ASD as they could predict the presence of ASD with a 98% correct classification rate using Fisher discriminant analysis [18].

Impairments in the methylation cycle commonly lead to an accumulation in homocysteine, a finding which was reported in a number of studies in ASD [19,20,21], including a meta-analysis of 31 studies [22]. However, some studies also report lower homocysteine levels in some children with ASD [8,9]. This may be related to the fact that homocysteine is the precursor to two pathways. First, methionine synthase (MS), a B12 and folate-dependent enzyme, converts homocysteine to methionine; one study reported lower MS mRNA in the brains of individuals with ASD (especially in younger children) compared to controls [23]. Lower MS mRNA would presumably result in lower production of the MS enzyme, which would slow the conversion of homocysteine to methionine and result in a build-up of homocysteine. However, homocysteine is also the precursor to cysteine, which is the rate-limiting substrate for the production of GSH. ASD is associated with lowered concentrations of GSH in blood [8,9,24], mitochondria [25], lymphoblasts [25], and brain tissue [15,26], and a recent meta-analysis found consistent depletion of GSH in ASD across 14 studies [5]. Since GSH is produced from homocysteine (and cysteine), GSH depletion may lead to the depletion of homocysteine by consuming it as a precursor. Impaired methylation can also weaken sulphation pathways; lower blood and higher urinary sulfate concentrations have been reported in ASD as early as 1997 [27,28,29]. 

Treatments that have been shown to improve methylation and GSH production in ASD include methylcobalamin (mB12), betaine (anhydrous trimethylglycine), and leucovorin (folinic acid) [8,11]. Cobalamin (B12) exists in several forms: cyanocobalamin (cB12, a synthetic form of B12 not found in a natural form; available by injection or orally) and three naturally occurring forms: mB12, hydroxycobalamin (hB12), and adenosylcobalamin (aB12); the latter three natural forms have better bioavailability compared to cB12 and are available in an injectable form or orally [30]. B12 is important for brain development, and B12 deficiency has been associated with regression in social interaction in one child [31] and in another who developed Childhood Disintegration Disorder [32]. 

Although a number of studies have used B12 as a treatment in children with ASD, these studies have not been systematically reviewed to date. This systematic review identifies and collates the studies using B12 as a treatment in individuals with ASD. When possible, this review lists out the type(s) of B12 used along with dosing, route of administration, types of studies, clinical outcomes, and adverse events (AEs). Meta-analysis was used to examine biochemical changes in methylation and redox metabolism as well as AEs.

## 2. Materials and Methods

### 2.1. Search Process

A prospective protocol for this systematic review was developed a priori, and the search terms and selection criteria were chosen in an attempt to capture all pertinent publications. A computer-aided search of PUBMED, Google Scholar, EmBase, Scopus, and ERIC databases from inception through June 2021 was conducted to identify pertinent publications using the search terms ‘autism’, ‘autistic’, ‘Asperger’, ‘ASD’, ‘pervasive’, and ‘pervasive developmental disorder’ in all combinations with the terms “MB12”, “Methylcobalamin”, “Cobalamin”, “B12”, “Cyanocobalamin”, “Hydroxycobalamin”, “Adenosylcobalamin”, and “Vitamin B12.” References cited in identified publications were also searched to locate additional studies. 

### 2.2. Study Selection and Assessment

This systematic review and meta-analysis followed PRISMA guidelines [33]. The PRISMA Checklist is found in Appendix A, and the PRISMA Flowchart is displayed as Figure 2. One reviewer screened titles and abstracts of all potentially relevant publications. Studies were initially included if they (1) involved individuals with ASD; and (2) reported on the use of B12 as a treatment in ASD. Articles were excluded if they: (1) did not involve humans (for example, cellular or animal models); and or (2) did not present new or unique data (such as review articles or letters to the editor). After screening all records, 17 publications met inclusion criteria (see Figure 2); two reviewers then independently reviewed these articles for inclusion and assessed factors such as the risk of bias. As per standardized guidelines [34], selection, performance detection, attrition, and reporting biases were considered. Two DBPC studies that used only injected mB12 were identified, but since only one of these studies provided detailed descriptive statistics of the clinical outcome measures [35], a meta-analysis could not be conducted on clinical outcome measures. 

### 2.3. Meta-Analysis

MetaXL Version 5.2 (EpiGear International Pty Ltd., Sunrise Beach, Queensland, Australia) was used with Microsoft Excel Version 16.0.12827.20200 (Redmond, WA, USA) to perform the meta-analysis. The data from this meta-analysis is available upon request to the authors. Random-effects models, which assume variability in effects from both sampling error and study level differences [36,37], were used to calculate incidence across studies (AEs Meta-analysis) while pooled Cohen’s d’ (a measure of effect size) was calculated from the standardized mean difference of outcome measures using the inverse variance heterogeneity model since it has been shown to resolve issues with underestimation of the statistical error and spuriously overconfident estimates with the random effects model when analyzing continuous outcome measures (Biochemistry Meta-analysis) [38]. Effect sizes were considered small if Cohen’s d’ was 0.2; medium if Cohen’s d’ was 0.5; and large if Cohen’s d’ was 0.8 or higher [39]. Cochran’s Q was calculated to determine heterogeneity of effects across studies, and when significant, the I^2^ statistic (Heterogeneity Index) was calculated to determine the percentage of variation across studies that was due to heterogeneity rather than chance [40,41], and the Luis Furuya-Kanamori (LFK) Index derived from Doi plots was reviewed for significant asymmetries (>±2) [42,43].

## 3. Results

This section will discuss the type(s), routes, and dosing parameters of B12 used along with the type of study and phenotypes of patients (Section 3.1), biochemical changes (Section 3.2), clinical outcome measures (Section 3.3), and AEs (Section 3.4) associated with B12 treatment studies. In the clinical outcomes section, the outcomes are presented and organized by study type. 

### 3.1. B12 Administration: Type of B12, Route, Dosing, and Type of Study 

#### 3.1.1. Type of B12

Five studies did not specify the type of B12 used [32,44,45,46,47]. Two studies used oral cB12 [48,49]. One study used a 50/50 oral mixture of mB12 and cB12 [50]. One study initially used cB12 injections and switched to mB12 injections due to worsening anemia from cB12 in 2 children and also used oral hB12 in one child [51]. The remaining 8 studies used mB12 [8,11,35,52,53,54,55,56]. Thus, of the 12 studies specifying the type of B12 used, 10 (83%) used mB12 by itself or in combination with cB12.

#### 3.1.2. Route Parameters of B12

Two studies did not specify the route of B12 administration [45,52]. Six studies used subcutaneously (SC) injected mB12 [8,11,35,53,55,56] and 4 studies used intramuscularly injected B12 [32,46,47,51]. Five studies used oral B12 [44,48,49,50,54]. One study used both injected cB12 and oral hB12 [51]. Thus, of the 15 studies specifying the route of administration, 10 (67%) used an injected form.

#### 3.1.3. Dosing of B12

Six studies (including two DBPC studies) used subcutaneous mB12 injections at a dose ranging from 64.5 to 75 µg/kg/dose [8,11,35,53,55,56]. One study used a lower dose of 25–30 µg/kg/dose (up to 1500 µg), but the route of administration was not specified [52]. One study used oral mB12 at a dose of 500 µg per day [54] while 2 studies used oral cB12 in doses of 500–1600 µg per day along with a multivitamin/mineral supplement (MVI) [48,49]. Another study used a 50/50 mixture of mB12 and cB12 500 µg per day combined in an MVI [50]. One study used oral B12 1.2 µg per day but did not specify the type of B12 given [44]. One study used 1000 µg of B12 (type not specified) intramuscularly for 5 days and then weekly for 8 weeks [32]. One study used mB12 intramuscularly at 1 mg once per week in 2 children and 10 mg of oral hB12 in another child [51]. Finally, 3 studies did not list the dose of B12 given [45,46,47].

#### 3.1.4. Types of B12 Studied

Of the 17 treatments studies, 2 were DBPC studies using mB12 injections alone [35,55], 2 were DBPC studies using oral cB12 in a MVI preparation [48,49], 1 was a prospective, controlled study [50], 6 were prospective, uncontrolled studies [8,11,44,45,52,53], and 6 were retrospective case series/reports [32,46,47,51,54,56]. Two studies reported on the same cohort of patients [11,53].

#### 3.1.5. Phenotypes of Patients in B12 Studies

Table 1 lists the phenotypes for the 17 studies. Six studies used DSM-4 criteria to diagnose ASD [8,11,44,52,55,56]. Six studies used autism diagnostic observation schedule (ADOS) and/or autism diagnostic interview (ADI) to confirm the diagnosis [35,45,50,51,54,55]. Five studies did not specify criteria for autism [32,46,47,48,49]. Two studies only enrolled patients with abnormal biochemical findings (such as methylation abnormalities or oxidative stress) [11,53].

Autism Diagnostic Interview (ADI); Autism Diagnostic Observation Schedule (ADOS); Childhood Autism Rating Scale (CARS); childhood disintegrative disorder (CDD); Diagnostic and Statistical Manual of Mental Disorders, fourth edition (DSM-4); Gilliam Autism Scale (GARS); glutathione (GSH); Preschool Language scale, Fourth Edition (PSL4); oxidized glutathione (GSSG); Pervasive Developmental Disorder, Not Otherwise Specified (PDD-NOS); S-adenosylhomocysteine (SAH); S-adenosylmethionine (SAM); years old (yo).

### 3.2. Biochemical Changes with B12 Treatment

#### 3.2.1. Review of Studies

Eleven studies examined biochemical changes with B12 treatment and are outlined in Table 2. A 12-week randomized DBPC crossover study of 30 children with ASD (ages 3–8 yo) used mB12 64.5 µg/kg subcutaneous injections every 3 days for 6 weeks or a placebo. GSH and GSH/GSSG were measured before and after treatment. No significant changes in GSH related metabolites were observed comparing the treatment group to the placebo group. However, significant improvements in GSH and the GSH redox ratio were found in a subgroup of 9 children who were considered “responders” based on significant improvements on the CGI and 2 other behavioral outcomes, suggesting these may be biomarkers for identifying children who respond to mB12 treatment [55]. In another randomized DBPC study, 57 children with ASD received 75 µg/kg mB12 subcutaneous injections every 3 days for 8 weeks or placebo injections, and clinical improvements were positively associated with increased plasma methionine, decreased SAH, and improved methylation capacity as measured by the SAM/SAH ratio [35].

In a prospective study of 8 children with ASD, 75 µg/kg mB12 SC two times per week for one month was given along with folinic acid (800 µg twice a day orally) and betaine (1000 mg twice a day orally). This treatment led to significant decreases in adenosine and GSSG as well as significantly increased levels of methionine, cysteine, GSH, SAM/SAH, and GSH/GSSG [8].

In a study of 40 children with ASD, 75 µg/kg mB12 SC 2 times per week for 3 months given with folinic acid (400 µg twice per day) led to increased cysteine, cysteinylglycine, and GSH, and decreased GSSG [11]. Improvements in GSH redox status were associated with improvements in expressive communication, personal and domestic daily living skills, and interpersonal, play-leisure, and coping social skills [53]. 

In a case series of 3 patients with ASD with transcobalamin deficiency/transcobalamin II (TCN2) mutations (one patient had metabolic acidosis and pancytopenia), cB12 intramuscular injections (1 mg once or twice weekly) were started along with carnitine. However, both patients developed acute anemia on cB12 injections, which improved when switching to mB12 intramuscular injections 1 mg weekly; in the third child, homocysteine levels normalized with 10 mg per day of hB12 orally once per day [51].

In a randomized DBPC study of 141 children and adults with ASD, a MVI containing 500 mcg of oral cB12 led to significant improvements compared to placebo in total sulfate, SAM, reduced glutathione, GSSG:GSH, nitrotyrosine, adenosine triphosphate (ATP), NADH, and NADPH [49].

In a study of 13 patients with ASD, mB12 25–30 µg/kg/day (up to 1500 µg/day; route of administration not specified) was given for 6–25 months. Five patients had normal B12 serum levels and after the study, 4 cases had above normal serum B12 without any apparent AEs [52].

In another study of 30 children with ASD, oral B12 (type not specified) 1.2 µg per day given with vitamin B6 200 mg and folic acid 400 µg led to a significant reduction in urinary homocysteine [44]. In a prospective, uncontrolled study of 127 children with ASD, B12 treatment (type and route of administration not specified) was administered to 38% of the ASD group along with mitochondrial supplements. Analysis showed that B12 treatment improved complex I activity compared to not supplementing with B12. In addition, B12 treatment was associated with the better coupling of Complex I and Citrate Synthase mitochondrial enzymes [45].

Finally, one retrospective study examined 24 children with ASD (mean age 9.3 ± 3.5 years) who received SC mB12 injections (75 µg/kg, given every 1–3 days) and compared urinary and plasma cobalt levels to 48 children (mean age 8.9 ± 3.7 years) who did not receive mB12 injections. The mean plasma cobalt concentration in the mB12 group was 0.82 ± 0.19 µg/L compared to 0.12 ± 0.10 in the untreated group (*p* < 0.001). The investigators noted that the study was limited as it could not determine what form of cobalt was present (free or bound in mB12), how the cobalt was distributed in the body, and what the tissue levels would be. This study did not report clinical outcomes [56].

#### 3.2.2. Meta-Analysis of Biochemical Changes Related to Methylcobalamin

Only three studies [8,11,35] provided enough detailed information about biochemical metabolites before and after treatment to be included in a meta-analysis. All studies used subcutaneously injected mB12. One study [35] was a DBPC study that used mB12 alone, and two studies [8,11] were uncontrolled prospective studies, which used mB12 with additional treatments. Of these latter two studies, one study [8] obtained biochemical measurements three months after starting daily 800 μg of leucovorin (folinic acid) and 1000 mg of betaine and then three months after adding mB12. In the meta-analysis, the biochemical measurements directly before and after adding mB12 were used rather than using the baseline measures obtained before any treatments were added. For the second study [11], mB12 was provided along with daily 800 μg of leucovorin (folinic acid). Common metabolites across all three studies were analyzed, and the results are outlined in Table 3. Even though these 3 studies used different dosing parameters and two used mB12 with folinic and one without, both mB12 and folinic have been shown to lead to biochemical changes in methylation metabolites and work in a synergistic fashion. Therefore, combining these studies to examine biochemical changes was felt to be advantageous.

Meta-analysis of the methylation metabolites demonstrated small effect sizes that were in the direction expected for improvement of methylation metabolism for the majority of the metabolites. Specifically, the weighted mean differences suggested an increase in Methionine, SAM, and SAM/SAH ratio, and a decrease in SAH. None of the effects were overall statistically significant, and the Cochran’s Q statistic suggested that the variation was not due to heterogeneity across studies but rather due to random chance. However, for homocysteine, although the overall effect was not significant, the variation in the measurement was found to be largely due to variation across studies with a slight asymmetry in the Doi plots. This was due to one study [35] demonstrating a decrease in homocysteine with treatment and two studies [8,11] demonstrating an increase in homocysteine with treatment. 

Meta-analysis of the redox metabolism metabolites demonstrated statistically significant medium to large effect sizes that were in the direction expected for an improvement in redox metabolism for all of the metabolites examined. Specifically, the weighted mean differences demonstrated an increase in cysteine, total GSH, and total GSH/GSSG redox ratio, and a decrease in GSSG. The I^2^ statistic suggested that the variation was due to heterogeneity across studies, specifically one study [35] demonstrating more marginal effects as compared to the other two studies [8,11]. However, Doi plots did not demonstrate any significant asymmetries. 

### 3.3. Clinical Outcomes with B12 Treatment

Twelve studies examined clinical outcomes and are outlined in Table 4. Improvements reported in these studies included sleep, hyperactivity, gastrointestinal problems, tantrums, nonverbal IQ, expressive, written and receptive communication skills, daily living skills, social skills domains, vision, eye contact, echolalia, stereotypy, anemia, and nocturnal enuresis.

#### 3.3.1. Double Blind Placebo-Controlled Studies

Four DBPC studies examined B12 use in ASD, with two of the studies administering it alone as an injection in the form of mB12 and two studies using cB12 in an orally administered MVI.

In a 12-week randomized DBPC crossover study, 30 children with ASD (ages 3–8 yo) received mB12 64.5 µg/kg SC injections every 3 days for 6 weeks or placebo injections for 6 weeks; a 6-month extension study was also performed. No significant changes in clinical outcomes were observed comparing the treatment group to the placebo group. A total of 22 children entered the 6-month open-label extension portion of the study, but no clinical outcomes were reported for this time period in the publication. A subgroup of 9 children were considered “responders” based on significant improvements on the CGI and 2 other behavioral outcomes. The authors commented that the crossover design without a washout period between the two treatments might have made it more difficult to see a significant difference between groups [55]. In another randomized DBPC study by some of the same investigators, 57 children with ASD received 75 µg/kg mB12 subcutaneous injections every 3 days for 8 weeks or placebo injections. A significant improvement in CGI-I as rated by clinicians was observed with mB12 treatment, but no significant improvements in the treatment group were observed in the parent-rated aberrant behavior checklist (ABC) or social responsiveness scale (SRS). Notably, this study did not include the crossover design, which was felt to be a weakness in design from the first study [35]. 

In a randomized DBPC 3-month study of 20 children with ASD (ages 3–8 yo), a MVI, which also contained 1200–1600 mcg of oral cB12, led to significant improvements in sleep and gastrointestinal problems compared to placebo [48]. In another randomized DBPC study of 141 individuals with ASD, a MVI containing 500 mcg of oral cB12 led to significant improvements in several behavioral scales compared to placebo, including hyperactivity, tantrums, and receptive language [49].

#### 3.3.2. Prospective, Controlled Study

In a prospective, controlled, single-blind (clinical evaluators blinded) 12-month study, 37 individuals with ASD were treated with an oral MVI containing 500 mcg of B12 (50% as mB12 and 50% as cB12) along with other nutritional and dietary interventions and were compared to 30 control ASD individuals who were not treated. Significant improvements were observed in nonverbal IQ, communication, daily living skills, and social skills domains in the treated ASD group compared to controls [50].

#### 3.3.3. Prospective, Uncontrolled Studies

There were six prospective, uncontrolled studies of B12 in ASD [8,11,44,45,52,53], but no clinical outcomes were reported in four of these studies [8,11,44,45]. 

In a study of 13 patients with ASD, mB12 25–30 µg/kg/day (up to 1500 µg/day, route of administration unknown) given for 6–25 months led to significant improvements in IQ, developmental quotient, and Childhood Autism Rating Scale (CARS) score; improvements were similar in older children as in younger children, and in children with lower IQ compared to those with higher IQ [52].

In a study of 40 children with ASD, 75 µg/kg mB12 SC 2 times per week for 3 months given with folinic acid (400 µg twice per day) led to significant improvements on all subscales of the Vineland Adaptive Behavior Scale (VABS), including significant improvements in receptive, expressive, and written communication skills, personal, domestic, and community daily living skills, and interpersonal, play-leisure, and coping social skills. The average effect size of improvement was 0.59 (a moderate effect size), and the average improvement in skills development was 7.7 months [53]. This improvement over a 3-month period was consistent with the notion that the children started “catching up” in development. 

#### 3.3.4. Retrospective Studies

Five retrospective studies examined B12 treatment in ASD [32,46,47,51,54]. The first three studies used intramuscular B12 (type not specified) to treat a B12 deficiency. Improvements in vision were found in a case series of 3 children with ASD and optic nerve atrophy [46]. In a 9-year-old child with ASD, pulmonary hypertension and nutritional deficiencies (including B12 deficiency), B12 along with other nutritional supplements led to improvements in pulmonary hypertension and musculoskeletal problems [47]. Improvements in eye contact, licking fingers, hyperactivity, pacing, echolalia, repetitive behaviors, and in the CARS score were observed in a 14-year-old vegetarian boy with Childhood Disintegrative Disorder (CDD) with daily injections of 1000 µg B12 for 5 days and then weekly for 8 weeks along with oral antioxidants and vitamins [32]. In another study, daily oral mB12 500 µg resolved nocturnal enuresis a 18-year-old child with ASD, with the enuresis reoccurring when mB12 was stopped [54]. Finally, in one child, “developmental milestones improved” with 10 mg per day of hB12 orally once per day [51].

### 3.4. Adverse Effects of B12

Here the potential AEs (see Table 4) of injected and oral forms of B12 are discussed separately, followed by the studies in which the route of administration is unknown.

Both DBPC studies [35,55] that used mB12 injections reported AEs. No serious AEs were reported in either study. Hyperactivity and increased mouthing of items were reported as AEs in one study [55], but the study did not indicate whether these AEs were significantly different between the treatment and placebo groups or what percentage of children had one of these side effects. The other DBPC study [35] reported 21 adverse events in the mB12 group compared to 24 in the placebo group (no significant difference). Adverse events in the mB12 group included a cold (11%), fever (7%), flu (4%), growing pains (4%), increased hyperactivity (7%), increased irritability (4%), lack of focus (4%), mouthing (19%), nosebleed (7%), rash (4%), stomach flu (4%), and trouble sleeping (4%); these were not significantly different compared to the placebo group. Two prospective, uncontrolled studies used mB12 injections [8,53], with one reporting AEs [53]. Adverse events in this study included hyperactivity (10%, improved when the folinic acid dose was lowered), sleep disruption (3%), difficulty falling asleep (3%), increased impulsiveness (3%), and irritability (3%). After study completion, 31/40 (78%) of parents indicated a desire to continue treatment (8 parents (20%) did not respond to this question) [11,53]. Of the five case reports using injected B12, only two reported AEs. Two patients who received cB12 injection developed acute anemia, which resolved by changing the cB12 form to mB12 injections [51]. In a child receiving oral mB12, increased motor stereotypy and hyperactivity were noted within 100 days of starting treatment [54]. Finally, one retrospective study reported higher plasma and urinary cobalt levels in patients who received SC mB12 compared to controls, but no apparent adverse events from this finding were observed [56].

Of the two DBPC studies that used B12 incorporated into an oral MVI [48,49], one reported no side effects in patients who followed the correct dosing parameters, but nausea and vomiting in 2 children (18%) in the treatment group when the MVI was taken on an empty stomach (against study protocol) [48]. The other study reported aggression (4%), night terrors (2%), trouble focusing (2%), moodiness (2%), nausea (2%), diarrhea (2%), mild behavioral problems (11% compared to 7% in the placebo group, *p* = ns), and diarrhea/constipation (11% compared to 7% in the placebo group, *p* = ns) with 2 participants withdrawing due to aggression and one withdrawing because of nausea and diarrhea [49]. In a prospective controlled single-blind study using an oral MVI, which included a 50/50 mixture of cB12 and mB12, AEs included moderate worsening of behaviors in 2 children (6%) found to have low levels of nutrients, particularly extremely low levels of cobalamin [50]. 

Three retrospective case series using intramuscular cobalamin with the type of B12 not specified did not report if any side effects occurred [32,46,47]. Two studies did not report the route of administration. One prospective uncontrolled study using mB12 (route not specified) reported there were no significant adverse events [52]. Another study did not examine AEs specifically [45]. 

#### Meta-analysis of Adverse Effects Related to Cobalamin

To better understand the AEs associated with cobalamin treatment, a meta-analysis was performed on the reported AEs across studies for studies using injected and oral B12 separately (see Table 5). Case reports were excluded from this analysis due to the potential bias of reporting in such studies, and only studies that included quantitative measures of the AEs in the samples studied were included. For injected B12, only two studies [11,35] met this criterion, with both studies using subcutaneously injected mB12. For oral B12, three studies [48,49,50] met this criterion, all using an MVI including B12 in various forms. 

There was a low incidence (<5%) of most AEs for subcutaneously injected mB12 with only three AEs reaching significance: increased irritability (3.4%), trouble sleeping (7.6%) and increased hyperactivity (11.9%), suggesting that for most individuals with ASD, subcutaneously injected mB12 is well tolerated without AEs. There was also a low incidence (<5%) of most AEs for B12 included in an oral MVI. Unlike injected mB12, the oral route was associated with gastrointestinal AEs, although these were also at a low incidence (<5%). Significant AEs for oral formulation included aggression (1.8%) and worsening behavior (7.7%), which is similar in incidence and character to the significant behavioral AEs seen in the injected route. However, again, most participants tolerated the B12 in a MVI without any AEs, suggesting that, in general, it was well tolerated. 

## 4. Discussion

This review identified 17 studies using B12 as a treatment for ASD. Table 6 summarizes the major findings of these studies. Two studies were DBPC controlled studies using mB12 injections and two were DBPC studies using cB12 combined in an MVI. Most studies that specified the type of B12 used mB12 (10/12, 83%). Overall, the treatment was well tolerated with minimal AEs, and the majority of the studies reported positive effects on ASD symptoms, although some studies found that these improvements were limited to a subgroup of children with baseline unfavorable biochemistry. 

### 4.1. Improvements in Autism Symptoms with Cobalamin

Cobalamin treatment was found to result in improvements in both core and associated ASD symptoms. Three of the prospective studies examined changes in biochemistry and gave subcutaneously injected mB12 with two of the studies demonstrating overall clinical improvements and all studies suggesting improvements in a subgroup with unfavorable biochemistry. The study that did not show an overall effect used a DBPC crossover design but not a washout period between the crossover, potentially making it more difficult to observe a significant difference between treatment groups [55]. The other DBPC study did not use a crossover design and was able to document improvements as ranked by clinicians compared to placebo [35]. The other prospective study was open-label but differed in two critical ways from the others in its design: first, subcutaneous mB12 was combined with oral folinic acid, and second, only participants with unfavorable biochemical profiles consisting of decreased methylation capacity and decreased GSH redox ratio were entered into the study. This latter study demonstrated significant improvements on all subscales of the VABS with a moderate effect size (0.59) and an average improvement in skills development of 7.7 months during the three-month treatment period. This suggests that mB12 treatment with folinic acid might help some children with ASD “catch up” in development [53]. Other retrospective studies of injected B12 are consistent with these prospective studies, demonstrating improvement in physical health as well as core (e.g., eye contact, echolalia, and repetitive behaviors) and associated (e.g., hyperactivity) ASD behaviors [32,46,47]. 

Oral B12 combined with a MVI has been studied in several prospective controlled studies with these studies demonstrating improvements in physical medical issues such as gastrointestinal problems and sleep [48], as well as in behaviors such as hyperactivity and tantrums [49], and in cognitive skills, including language [49], non-verbal IQ and social skills [50]. Oral mB12 alone improved nocturnal enuresis in one study [54]. One study in which the route of treatment was not known found that mB12 gave similar improvements in both older and younger children as well as those with lower IQ. This suggests that even older and more severely affected individuals with ASD can improve with mB12 treatment [52].

### 4.2. Biochemical Effects of Cobalamin

Studies have reported a wide variety of improvements in biochemistry associated with B12 treatment, including improvements in methylation with injected [8,35] and oral [44,49] B12, redox metabolism with injected [8,11] and oral [49] B12 and mitochondrial function with oral [49] or unspecified [45] B12 treatment. 

The meta-analysis examined whether there were consistent changes in biochemistry with B12 treatment. Changes in methylation were not found to be consistent when combining studies. However, homocysteine at baseline was markedly lower in the studies that demonstrated an increase in homocysteine with treatment (baseline homocysteine 6.7 ± 0.7 [8] and 4.8 ± 1.8 [11]) as compared to the study that demonstrated a decrease in homocysteine (baseline homocysteine 8.9 ± 1.0 [35]). Furthermore, even after treatment, in the studies where homocysteine started out lower, the post-treatment homocysteine concentrations were still lower than the post-treatment homocysteine in the study where homocysteine started out higher. This was most likely driven by differences in the design of the studies. In the two studies in which homocysteine was low, entry into the study required an unfavorable biochemistry profile [8,11], whereas, in the study with the higher homocysteine, no such criterion was implemented [35]. This may reflect that mB12 can drive homocysteine to an optimal concentration with the direction of change dependent on the starting concentration for the particular participant. Clearly, larger studies, which are sensitive to the baseline biochemical abnormalities, are needed to examine the complexity of the effect of mB12 on methylation. The meta-analysis did demonstrate significant improvements in transsulfuration and redox metabolism across studies with a medium to large effect size, suggesting that mB12 may be important for improving redox metabolism independent of the abnormalities in methylation metabolism.

The strongest evidence that the effect of B12 on biochemistry is clinically important is reflected in the studies which demonstrate that changes in biochemistry are associated with positive changes in clinical symptoms. Although one DBPC study of subcutaneously injected mB12 did not find an overall effect of mB12 on clinical outcomes, it did find that a subgroup of clinical “responders” showed improvements on the clinical global impression scale and at least two additional behavioral measures; these responders had significant improvements in GSH and the GSH redox ratio, suggesting these biomarkers could be used to predict response to mB12 treatment in children with ASD [55]. The second DBPC study of subcutaneously injected mB12 demonstrated that clinician-rated improvements were significantly and positively associated with improvements in methylation metabolism [35]. Finally, in another prospective study of subcutaneously injected mB12, significant improvements in the GSH redox ratio were significantly associated with clinical improvements, including expressive communication, personal and domestic daily living skills, and interpersonal, play-leisure, and coping social skills [53]. These findings further support the notion that certain biochemical abnormalities might predict treatment response to mB12 injections. 

### 4.3. Biological Mechanisms of Actions

There are several potential biological mechanisms of action of B12 treatment. First, as demonstrated by the biochemical meta-analysis, subcutaneously injected mB12 consistently and significantly improves redox metabolism, including increasing Cysteine, GSH, and the GSH redox ratio as well as decreasing GSSG with medium to large effect sizes. A systematic review and meta-analysis have demonstrated that individuals with ASD overall demonstrate low GSH and Cysteine and increased GSSG across multiple research groups in multiple countries [57]. These redox metabolism abnormalities have been documented in multiple tissues in individuals with ASD, including in post-mortem brain [15,58]. In addition, these redox abnormalities have been shown to result in oxidative damage to DNA, protein, and lipids, as well as mitochondrial dysfunction and inflammation in the brain of individuals with ASD [15,59]. Thus, improving redox abnormalities can have widespread positive effects on the physiological function of the brain and other key important organs such as the immune system and GI tract in individuals with ASD. 

Second, the brain of individuals with ASD has been repeatedly shown to have an excitatory/inhibitory imbalance such that the cortex is overexcited and underinhibited [60]. Glutamate is the major excitatory neurotransmitter of the cortex, thus decreasing glutamate neurotransmission is believed to be potentially therapeutic in ASD [61]. Glutamate is one of the three key precursors of GSH, along with cysteine and glycine. Thus, improved production of GSH, in part driven by mB12, may have other positive effects in neurotransmitter metabolism by reducing glutamate concentrations in the cortex (See Figure 3).

Third, polymorphisms in TCN2, the cobalamin binding protein, are associated with an increased risk of ASD [9], and ASD is one of the characteristics of some individuals with a mutation in TCN2 [51]. Thus, higher concentrations of B12 may be required in the blood in those with defects in the cobalamin binding protein in order to obtain sufficient levels of B12 into the tissues, including the brain.

Fourth, one controlled study reported a more than 3-fold reduction in mB12 concentration in brain tissue from individuals with ASD [58]. The reason for this finding is not clear but it is very possible that, such as the case with folate, some individuals with ASD may have a defect in the transport mechanism for B12 into the brain. Indeed, in one study, it was found that the folate receptor alpha autoantibody was associated with higher levels of blood cobalamin concentrations [62]. These higher blood concentrations of B12 could potentially be caused by B12 being blocked from uptake into the tissues. If such a transportation mechanism is present, a high blood concentration of B12 may be needed to overcome the blockage in the transportation mechanism, similar to the use of high dose folinic acid as a therapy for dysfunction of the folate receptor alpha [63]. Notably, one study reported that higher than normal serum concentrations of B12 was not associated with any observable adverse effects in individuals with ASD [52].

Fifth, methionine synthase, the key B12 dependent enzyme required for the proper functioning of the methylation cycle may be dysfunctional in the brain of individuals with ASD as one study observed that the mRNA for this enzyme is underexpressed in brain tissue of individuals with ASD [23]. Thus, high concentrations of B12 may be important to allow the limited amount of this enzyme to function optimally in the brain.

Sixth, B12 could be helpful in improving mitochondrial function. Two studies reported improvements in mitochondrial-related markers in children with ASD. One prospective, controlled, single-blind 12-month study reported improvements in mitochondrial related markers including ATP, NADH, and NADPH with an oral MVI containing 500 mcg of B12 (50% as mB12 and 50% as cB12) along with other nutritional treatments [49]. In another prospective study in 127 children with ASD, B12 treatment was associated with the better coupling of Complex I and Citrate Synthase mitochondrial enzymes [45]. Since mitochondrial dysfunction is a common medical comorbidity in children with ASD [3,4], the use of B12 appears important in treating this problem.

Lastly, some individuals with ASD have lower intake and blood levels of B12 compared to controls. For example, a meta-analysis of 29 studies reported children with ASD had a significantly lower dietary intake of B12 compared to controls [64], while another meta-analysis of 16 studies found plasma B12 concentrations significantly lower in ASD compared to controls, although evidence of potential publication bias was found [5]. These finding may be due to feeding difficulties, which are common in individuals with ASD [65]. Thus, B12 treatment may be correcting a deficiency in some children with ASD. In addition, because the absorption of B12 orally may be dependent on adequate calcium intake [66], future studies examining the effect of calcium intake in conjunction with B12 might be helpful in ASD. Since some patients with ASD take multiple nutritional supplements, the potential interaction between B12 and other nutritional supplements also warrants further studies.

### 4.4. Formulation

Most studies (67%) that specified the route of administration used an injected form of B12. All studies that used the injected formulation and specified the type of B12 used mB12, except for one study, which initially used cB12 and changed to mB12. All studies that have linked changes in biochemistry with improvements in autism symptoms used subcutaneously injected mB12. The injected formulation is preferred by some practitioners because individuals with ASD may not be able to successfully take B12 orally due to GI disorders. Indeed, gastritis and enterocolitis may prevent optimal absorption while sensory aversions or esophagitis may prevent ASD children from easily swallowing a supplement [67,68]. Furthermore, as mentioned above, if there is a defect in B12 transportation in the blood or across the blood–brain barrier, higher blood concentrations of B12 may be needed to optimally deliver B12 into the organs, particularly the brain. It is possible that the GI tract is not even designed to absorb the necessary quantity of B12 that might be needed if indeed a problem with B12 transportation is present. Injected B12 may help raise the blood level higher than that achieved with oral B12.

A few studies combined B12 with other vitamins and/or minerals, thus these studies may reflect the combination of B12 with other supportive treatments. One study examined changes in biochemical markers of methylation and redox metabolism before and after adding betaine and folinic acid, and then after adding subcutaneously injected mB12 and found that the addition of mB12 to betaine and folinic acid both had a positive effect on methylation and redox metabolism [8]. Thus, in studies that have used other supportive treatments, B12 may have worked in concert with these other treatments. A recent randomized, single-blind study using a combination of various dietary and nutritional treatments in individuals with ASD, including essential fatty acids, Epsom salt baths, digestive enzymes, carnitine, and a MVI containing a combination of mB12 and cB12 reported significant improvements over one year compared to a control group that was not treated, suggesting a combination of treatments, which affect methylation and redox metabolism might lead to more robust improvements [50]. Further research will be needed to understand the optimal combination of treatments. 

### 4.5. Adverse Events

The meta-analysis of AEs demonstrated a low incidence of AEs with most children tolerating the treatment without any AEs. Only one study described a potentially serious AE, anemia when using injected cB12. This AE resolved with changing to injected mB12. mB12 given by injection or orally has not been associated with any severe or serious AEs, perhaps suggesting that it is the preferred form of B12 for individuals with ASD at this time. After the completion of one study, 31/40 (78%) of parents indicated a desire to continue mB12 treatment [11,53]. This suggests that parents observed enough clinical improvements and a low enough rate of adverse effects to continue mB12 injections. Interestingly, although the majority of significant AEs reported pertained to worsening behavior and sleep, some studies have demonstrated significant improvements in such symptoms, including hyperactivity [32,49] and sleep [48], suggesting that overall many children have improvements in these key symptoms, and that having such symptoms at baseline is not necessarily a contraindication for starting B12 treatment. One study reported some individuals with ASD had above normal serum B12 levels with B12 treatment without any apparent AEs [52]. Finally, one retrospective study examined 24 children with ASD who received mB12 injections and compared urinary and plasma cobalt levels to 48 children who did not receive mB12 injections. These children were treated with the standard dose of mB12 used in the SC mB12 studies (75 µg/kg, injected every 1–3 days). The mean plasma cobalt concentration in the mB12 group was significantly elevated (0.82 ± 0.19 µg/L) compared to the untreated group (0.12 ± 0.10). The investigators noted that the study was limited as it could not determine what form of cobalt was present (free or bound in mB12), how the cobalt was distributed in the body, and what the tissue levels would be. Furthermore, no apparent adverse events from this finding were reported in the study [56]. Notably, adverse effects from serum cobalt are unlikely to occur with a blood cobalt concentration under 100 µg/L (using a conservative estimate), which is over 100 times the level reported in this aforementioned study [69]. Furthermore, no study has reported AEs consistent with cobalt toxicity (e.g., iron deficiency, pernicious anemia, cardiomyopathy, and polycythemia).

### 4.6. Limitation of Published Studies

Many studies demonstrated detection bias as only a few studies used standardized outcomes. This made it impossible to perform a meta-analysis across clinical outcomes. Second, although some studies were prospective, they were uncontrolled and unblinded and not randomized, opening up the possibility of selection and performance bias. Finally, the retrospective studies had the potential drawback of being open to attrition and reporting bias. 

## 5. Conclusions

Overall, B12 appears to have evidence for effectiveness in individuals with ASD, particularly in those who have been identified with unfavorable biochemical profiles. In general, B12 appears to be very well-tolerated and safe. Two types of B12 have been studied in controlled and/or prospective clinical trials: (1) subcutaneously injected mB12 has evidence for improving clinical symptoms of ASD and improving methylation and redox metabolism, especially in those with unfavorable biochemistry and when combined with folinic acid (aka leucovorin) and (2) a mixture of cB12 and mB12 included in a MVI also appears to be associated with improvements in clinical symptoms, biochemistry, and physical medical disorders. 

As mentioned above, the current set of studies have their limitations and should be used to design and implement well-controlled blinded randomized clinical trials in the future. Additionally, the ASD population is very heterogeneous, making it important to understand the subset of children with ASD in which B12 treatment may be most effective. There is evidence that B12, particularly subcutaneously injected mB12, may be particularly helpful for a subset of individuals with ASD with unfavorable biomarkers, suggesting that biomarkers need to be studied alongside clinical outcomes with a prior hypothesis regarding subgroups that may optimally respond to treatment. Thus, including reliable biomarkers that can guide treatment will be helpful to optimize this potentially important well-tolerated safe treatment.

## Figures and Tables

**Figure 1 jpm-11-00784-f001:**
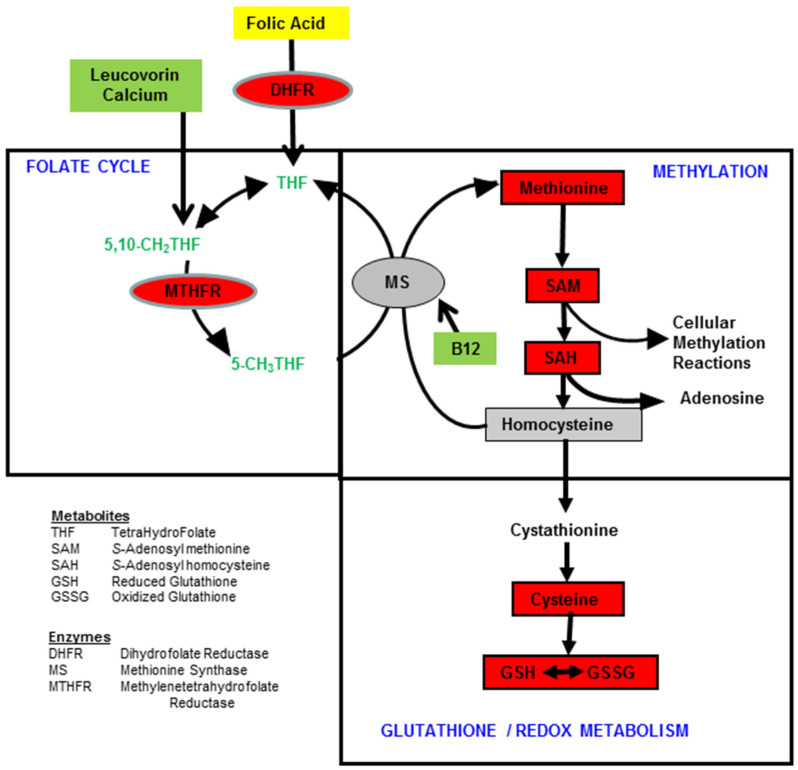
The connected folate, methylation, and redox cycles. Ovals represent enzymes and boxes represent metabolites. Red indicates metabolites and enzymes repeatedly noted to be consistently abnormal in ASD. Green highlights treatments that improve the metabolism of these cycles. Folic acid is shown in yellow as it is a suboptimal treatment because it is oxidized and has to be converted to active forms of folate.

**Figure 2 jpm-11-00784-f002:**
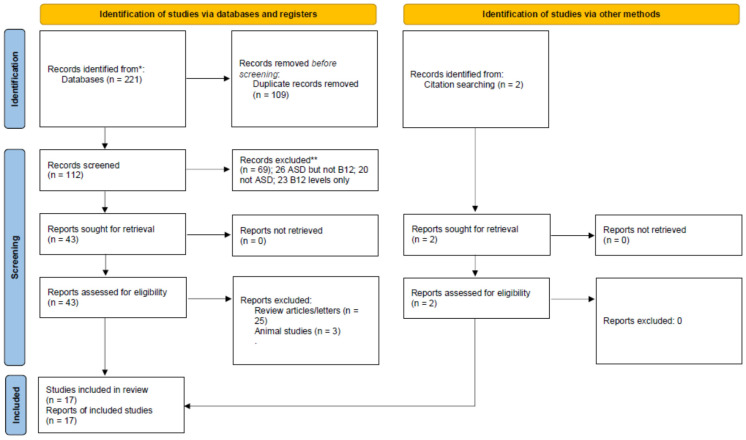
PRISMA 2020 flow diagram for this systematic review.

**Figure 3 jpm-11-00784-f003:**
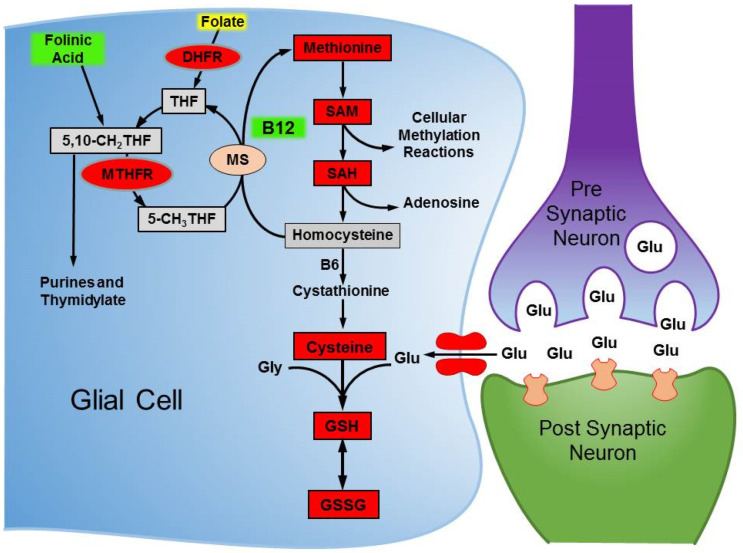
The production of glutathione results in the consumption of glutamate, the major excitatory neurotransmitter of the cortex, thus reducing glutamate in the brain. This diagram shows the connected folate, methylation, and redox metabolic pathways in the brain. Ovals represent enzymes and boxes represent metabolites. Red indicates metabolites and enzymes repeatedly noted to be consistently abnormal in ASD. Green highlights treatments that improve the metabolism of these cycles. Folic acid is shown in yellow as it is a suboptimal treatment because it is oxidized and has to be converted to active forms of folate.

**Table 1 jpm-11-00784-t001:** Phenotypes of autism in the 17 reviewed studies, by year published.

Study	Phenotype
Adams and Holloway, 2004 [48]	20 children with autism (age 3–8 yo); diagnosis of an autism spectrum disorder (autism, pervasive developmental disorder/not otherwise specified [PDD/NOS], or Asperger’s syndrome) by a psychiatrist or developmental pediatrician (criteria not specified); no changes in any treatments during the preceding two months; no previous multivitamin/mineral supplement use (except a standard children’s multivitamin/mineral)
James, et al., 2004 [8]	20 children with autism (mean age 6.4 yo, all white, 14 boys, 6 girls); 19 with regressive autism, 1 with infantile autism. Autism was based on the criteria for autistic disorder (DSM-4) by a developmental pediatrician; most had speech and socialization impairments and gastrointestinal problems
Nakano, et al., 2005 [52]	13 patients (2–18 yo, 11 male, 2 female) with autism, diagnosed by DSM-4 criteria by a pediatric neurologist; 4 with epilepsy and 1 with periventricular leukomalacia
James, et al., 2009 [11]	40 children with autism (2–7 yo, 33 boys, 7 girls) diagnosed by DSM-4 criteria and a CARS score >30; exclusion criteria included Asperger’s disorder, PDD-NOS, genetic disorders, seizures; severe gastrointestinal problems, recent infection, and use of high-dose vitamin or mineral supplements; all children had reduced methylation capacity (SAM:SAH) or reduced GSH redox ratio (GSH:GSSG) as an inclusion criterion
Bertoglio, et al., 2010 [55]	30 children with autism (3–8 yo, 28 boys, 2 girls) diagnosed by DSM-4 criteria and ADOS/ADI. No current use of mB12; nonverbal IQ of 49 or higher confirmed by a psychologist (Wechsler Preschool and Primary Scale of Intelligence, Mullen Scales of Early Learning, or the Wechsler Intelligence Scale for Children)
Geier and Geier, 2010 [56]	24 children with autism (mean age 9.3 yo, 23 white, 3 minorities, 21 boys, 3 girls) diagnosed by DSM-4 criteria
Pineles, et al., 2010 [46]	3 children with autism (6 yo boy, 13 yo boy, 7 yo boy) and vision loss/optic neuropathy (diagnostic criteria not specified)
Adams, et al., 2011 [49]	141 individuals with autism (3–60 yo, 125 boys, 16 girls); diagnosis of autism, PDD/NOS, or Asperger’s syndrome by psychiatrist or similar professional (criteria not specified); No vitamin/mineral supplement in the preceding 2 months
Kaluzna-Czaplinska, et al., 2011 [44]	30 children with autism (4–11 yo, 27 boys, 3 girls); autism diagnosed by DSM-4 criteria; all children on a sugar-free diet
Duvall, et al., 2013 [47]	9 yo child with ASD (criteria not specified)
Frye, et al., 2013 [53]	Same cohort as James et al., 2009 [11]
Malhotra, et al., 2013 [32]	14 yo boy with CDD (vegetarian)
Corejova, et al., 2015 [54]	18 yo adult male with autism diagnosed by ADI and CARS-2 by psychologist
Hendren, et al., 2016 [35]	57 children with autism (3–7 yo, 45 boys, 12 girls) with diagnosis confirmed by ADI and ADOS (the latter if needed) performed by a psychiatrist; IQ > 50 (Stanford–Binet 5th edition verbal subtest); excluded children with bleeding disorder, seizures, cancer, or genetic abnormalities, perinatal brain injury, current use of B12, or “other serious medical illness”
Delhey, et al., 2017 [45]	127 children with autism (ages 3–14 yo), diagnosis confirmed with ADOS and/or ADI, the state of Arkansas diagnostic standard, and/or DSM-4; prematurity was an exclusion criterion; 38 had mitochondrial function measured
Nashabat, et al., 2017 [51]	3 children with autism (3–2 yo, all boys); one diagnosed by GARS and PSL4; other 2 diagnosed by ADOS
Adams, et al., 2018 [50]	37 individuals (3–58 yo, 30 male, 7 female) with autism (29), PDD-NOS (3) or Asperger’s disorder (3); diagnosis confirmed by ADOS or CARS-2 by a psychiatrist, psychologist, or developmental pediatrician; no current use of nutritional supplements; patients with metabolic or genetic disorders not excluded; 13 had developmental regression, 15 with early-onset autism and 8 with developmental plateau; 9 with asthma; 13 with food allergies; 19 with other allergies

**Table 2 jpm-11-00784-t002:** Biochemical outcomes for treatment studies of cobalamin injections for autism spectrum disorder. Clinical global impression scale (CGI); subcutaneous (SC); years old (yo).

Study	Participants	Treatment	Outcomes
Subcutaneous Methylcobalamin Injections			
Bertoglio et al., 2010 [55] DBPC Crossover	30	64.5 µg/kg SC every 3 days	Significant improvements in GSH and the GSH redox ratio in a subgroup of 9 children who were considered “responders”
Hendren et al., 2016 [35] DBPC Parallel	57	75 µg/kg SC every 3 days	Clinical improvements were positively associated with increased plasma methionine, decreased SAH, and improved methylation compacity as measured by the SAM/SAH ratio
James et al., 2004 [8] Prospective Uncontrolled	8	75 µg/kg 2x/wk for 1 month with folinic acid and betaine	Decreased adenosine and GSSG; increased methionine, cysteine, tGSH, SAM/SAH, and tGSH/GSSG
James et al., 2009 [11] Prospective Uncontrolled	40	75 µg/kg 2x/wk for 3 months with folinic acid	Increased cysteine, cysteinylglycine, and GSH;decreased GSSG
Frye et al., 2013 [53] Prospective Uncontrolled			Clinical improvements associated with biochemical improvements in GSH redox status
Geier and Geier 2010 [56] Retrospective, controlled	24	75 µg/kg SC every 1–3 days	Urinary and plasma cobalt levels higher in mB12 group
**Intramuscular Methylcobalamin Injections**			
Nashabat et al., 2017 [51] Retrospective Uncontrolled		cB12 switched to mB12; one child received hB12	Improvement in anemia and metabolic acidosis when cB12 was changed to mB12; homocysteine normalized with 10 mg of hB12 orally in one child
**Oral Cobalamin with Other Vitamins**			
Adams et al., 2011 [49] DBPC Parallel	141	500mcg Daily Orally	Improvements in total sulfate, SAM, reduced GSH, GSSG:GSH, nitrotyrosine, ATP, NADH, and NADPH
Kaluzna-Czaolinska et al., 2011, [44] Prospective Uncontrolled	30	B12 1200 mcg per day with vitamin B6 200 mg and folic acid 400 µg	Reduction in urinary homocysteine
**Cobalamin with Route of Administration Not Specified**			
Delhey et al., 2017 [45] Prospective Uncontrolled	127	Any B12 taken in 38% of Patients along with other supplements	Improved coupling of Complex I and Citrate Synthase mitochondrial enzymes
Nakano et al., 2005 [52] Prospective Uncontrolled	13	mB12 25–30 µg/kg/day (up to 1500 µg) for 6–25 months	Increased B12 blood concentrations

**Table 3 jpm-11-00784-t003:** Meta-analysis of biochemical changes with subcutaneously injected methylcobalamin in children with ASD. Oxidized glutathione (GSSG); S-adenosylhomocysteine (SAH); S-adenosylmethionine (SAM); methylation capacity (SAM/SAH); total GSH (tGSH); total glutathione redox ratio (tGSH/GSSG) * *p* < 0.05, ** *p* < 0.01, *** *p* < 0.0001.

Metabolite	Weighted Mean Difference	Cohen’s d’ (95% CI)	Cochran’s Q	I^2^	LFK Index
Methylation Metabolites					
Methionine (μmol/L)	0.95	0.25 (−0.10, 0.59)	0.12		
SAM (nmol/L)	1.80	0.12 (−0.23, 0.47)	2.12		
SAH (nmol/L)	−0.78	−0.19 (−2.14, 0.58)	0.53		
SAM/SAH Ratio	0.22	0.28 (−0.07, 0.62)	0.66		
Homocysteine (μmol/L)	0.01	0.10 (−0.25, 0.45)	6.94 *	71%	1.03
**Transsulfuration/Redox Metabolites**					
Cysteine (μmol/L)	13.30	0.70 (0.34, 1.07) ***	15.13 **	87%	0.72
Total GSH (μmol/L)	0.36	0.43 (0.08. 0.79) *	8.73 **	77%	−0.16
GSSG (nmol/L)	−0.004	−0.68 (0.32, 1.04) ***	7.13 *	72%	0.61
Total GSH/GSSG Ratio	7.43	0.84 (0.47. 1.21) ***	18.37 ***	89%	0.88

**Table 4 jpm-11-00784-t004:** Studies of cobalamin injections for autism spectrum disorder with clinical outcomes. Childhood autism rating scale (CARS); clinical global impression scale (CGI); subcutaneous (SC); years old (yo).

Study	Participants	Treatment	Outcomes	Adverse Effects
Subcutaneous Methylcobalamin Injections: Prospective Double-Blind Placebo Controlled Studies				
Bertoglio et al., 2010 [55] Crossover	30	64.5 µg/kg SC every 3 days	No change in overall behavior	Hyperactivity; mouthing objects
Hendren et al., 2016 [35] Parallel	57	75 µg/kg SC every 3 days	Significant improvement in CGI	No significant difference between groups
**Subcutaneous Methylcobalamin Injections: Prospective Uncontrolled Study**				
Frye et al., 2013 [53]	40	75 µg/kg 2x/wk for 3 months with folinic acid	Overall improvement in development. Clinical improvement associated with biochemical improvements	Hyperactivity (10%), reduced sleep (6%), impulsiveness (3%)
**Intramuscular Cobalamin Injections with Type of B12 not specified: Retrospective Case Series/Reports**				
Pineles et al., 2010 [46]	3	B12 IM	Improvement in vision	None reported
Duvall et al., 2013 [47]	1	B12 IM with oral MVI	Improvements in pulmonary hypertension and musculoskeletal problems	None reported
Malhotra et al., 2013 [32]	1	B12 IM with oral MVI	Improvements in eye contact, licking fingers, hyperactivity, pacing, echolalia, and repetitive behaviors and in CARS score	None reported
**Oral Multivitamin, which Included Cobalamin: Prospective Double-Blind Placebo-Controlled Studies**				
Adams and Holloway 2004 [48] Parallel	30	1200–1600 mcg Daily Orally	Improved gastrointestinal problems and sleep	No adverse events in those adhering to the protocol
Adams et al., 2011 [49] Parallel	141	500 mcg Daily Orally	Improved hyperactivity, tantrums and receptive language	Mild behavioral problems, diarrhea, constipation
**Oral Multivitamin, which Included Cobalamin: Prospective Controlled Study**				
Adams et al., 2018 [50]	37	500 mcg Daily Orally	Improved non-verbal IQ, daily living skills and social skills	Worsened behaviors, gastrointestinal disturbance, facial rash, aggression, stereotypy
**Oral Methylcobalamin: Retrospective Case Report**				
Corejova et al., 2015 [54]	1	mB12 500 µg orally per day	Nocturnal enuresis resolved; reappeared when treatment was stopped	Increased stereotypy and hyperactivity
**Oral Hydroxycobalamin: Retrospective Case Report**				
Nashabat, et al., 2017 [51]	3	hB12 10 mg orally pr day	Improved developmental milestones	None
**Methylcobalamin with Route of Administration Not Specified: Prospective Uncontrolled Study**				
Nakano et al., 2005 [52]	13	25–30 µg/kg/day (up to 1500 µg) mB12 for 6–25 months	Improvements in IQ, developmental quotient and CARS score	No significant events

**Table 5 jpm-11-00784-t005:** Meta-analysis of adverse effects associated with cobalamin in children with ASD. Multivitamin (MVI) * *p* < 0.05. Adverse effects that were statistically significant are in bold and italic.

Injected mB12		MVI Including B12	
Adverse Effect	Incidence (95% CI)	Adverse Effect	Incidence (95% CI)
Cold Symptoms	4.5% (0.0%, 20.1%)	***Aggression***	***1.8% (0.1%, 5.2%)* ***
Fever	3.3% (0.0%, 13.7%)	Attention Problems	1.5% (0.0%, 4.1%)
Flu Symptoms	2.0% (0.0%, 7.4%)	Constipation/Diarrhea	4.1% (0.0%, 14.0%)
Growing Pains	2.0% (0.0%, 7.4%)	Moodiness	1.5% (0.0%, 4.1%)
***Increased Hyperactivity***	***11.9% (5.2%, 20.8%)* ***	Nausea/Vomiting	3.5% (0.0%, 11.0%)
***Increased Irritability***	***3.4% (0.2%, 9.3%)* ***	Night Terrors	1.5% (0.0%, 4.1%)
Lack of Focus	2.0% (0.0%, 7.4%)	***Worsening Behavior***	***7.7% (3.5%, 13.3%)* ***
Mouthing	6.8% (0.0%, 33.5%)		
Nosebleeds	3.3% (0.0%, 13.7%)		
Rash	2.0% (0.0%, 7.4%)		
Stomach Flu	2.0% (0.0%, 7.4%)		
***Trouble Sleeping***	***7.6% (2.3%, 15.2%)* ***		
Impulsivity	2.0% (0.0%, 7.4%)		

**Table 6 jpm-11-00784-t006:** Summary of studies on cobalamin treatment for ASD with clinical outcomes. Childhood Autism Rating Scale (CARS); Clinical Global Impression Scale (CGI); glutathione (GSH); oxidized glutathione (GSSG); intelligence quotient (IQ); multivitamin (MVI); S-adenosylhomocysteine (SAH); S-adenosylmethionine (SAM); methylation capacity (SAM/SAH); subcutaneous (SC); total GSH (tGSH); total glutathione redox ratio (tGSH/GSSG).

Total Studies	Total Participants	Treatment	Outcomes	Adverse Effects
**Prospective Double-Blind Placebo Controlled Studies: Subcutaneous Injected Methylcobalamin**				
2	87	64.5–75 µg/kg mB12 SC every 3 days	Significant improvement in clinician rated CGI and correlated with increases in methionine and decreases in SAH and improved SAM/SAH	Hyperactivity; mouthing objects
**Prospective Double-Blind Placebo Controlled Studies: Oral Cobalamin Combined in Multivitamin**				
2	161	MVI containing B12 orally	Improvements in gastrointestinal problems, sleep, hyperactivity, tantrums, and receptive language	Nausea, vomiting, aggression, night terrors, and diarrhea
**Prospective Controlled Studies: Oral Cobalamin Combined in Other Vitamin**				
1	67	B12 orally with other vitamins	Improved nonverbal IQ, communication, and daily living and social skills.	Worsened behavior
**Prospective Uncontrolled Studies: Methylcobalamin Injections or oral B12**				
6	218	25–75 µg/kg mB12 2–3x/week	Improved IQ, development and CARS scores. Improved GSH redox status associated with improved expressive communication, personal and domestic daily living skills, and interpersonal, play-leisure, and coping social skills. Improvements in adenosine, methionine, cysteine, tGSH, GSSG, SAM/SAH, tGSH/GSSG, coupling of Complex I and Citrate Synthase mitochondrial enzymes.	Hyperactivity, reduced sleep, and impulsiveness
**Retrospective Case Series/Reports: Intramuscular Cobalamin Injections with Type of B12 not specified or oral B12**				
6	32	IM B12 Different Forms	Improvements in vision, eye contact, pacing, echolalia, repetitive behaviors and nocturnal enuresis	Anemia on cB12 with improved on mB12

## Data Availability

All data are presented within the article.

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
