# Peer review of "The Effectiveness of Cobalamin (B12) Treatment for Autism Spectrum Disorder: A Systematic Review and Meta-Analysis"

_jpm, 2021, doi:10.3390/jpm11080784_

Round 1

Reviewer 1 Report

The stated goal of this manuscript is to provide a systematic review of 17 studies using B12 as a treatment for ASD; the authors also attempt 2 meta-analyses, one focused on biochemical changes subsequent to treatment and the other to an analysis of adverse effects.

Autism spectrum is at present identified entirely clinically and widely understood to be a collection of disorders, rather than a unified entity.  There is anecdotal evidence for the efficacy of B12, with or without other related vitamin cofactors, in the treatment of individual cases.  Accordingly, this is an important topic to review. 

Introduction:  could be more accessible to those not already well informed with respect to the topic.  Also, in my opinion, if the main goal of this paper is to review studies using B12 as a treatment, the findings of these studies should be presented in the main body of the paper, not in the introduction:  perhaps ¶ 5 and 6 of introduction should be moved to become 3.1 Summary of studies reviewed. 

A strength of the manuscript is the robust approach to conducting a systematic study and good enunciation of the steps that were taken to achieve this.

One significant concern with the design of the research relates to the meta-analysis of biochemical changes.  This was conducted on 3 studies, one with administration of B12 alone and two of B12 + folinic acid with or without betaine.   The extent to which it is appropriate to group these studies for an evaluation of biochemical outcome is not discussed or defended.

Throughout the text, many figures and tables precede, rather than follow, the first mention of same.  This significantly interrupts the flow of reading the communication.  Table 4 is not mentioned in the text at all. 

Overall, I find this manuscript to be lengthy and rather poorly organized.  Importantly, there is not description of the baseline phenotypes of the patients who did or did not respond to the treatment.  It may well be that this information was not available in the reviewed publications, but it is probably the single most important issue to explore with respect to evaluating the utility of this type of treatment.

Specific issues: 

Line 128:  Figure 2 does not list publications, but rather is the flowchart, as mentioned in line 132.  Table 5?  Is this a typo? 

Table 1 is labelled Biochemical changes…  but not all of the outcomes are biochemical in nature.  Table 1 refers to “methylation status” but the text refers to “methylation compacity [sic}.”  Methylation status will mean something different to molecular geneticists, so suggest that methylation capacity might be better.

Lines 231-235:  Patients with TCN2 mutations may have autistic symptoms, but they also have a genetically defined disorder, and should probably not be grouped with ASD patients not otherwise defined.

Discussion:  Lines 524-542:  not clear what the role of 2 additional studies might be here.  If the authors mean to discuss this as a possible mechanism underlying the effects of B12/cobalamin, they need to formulate this part of the discussion differently.

Review text for grammatical errors:  eg line 11 (2nd sentence of abstract): no verb; line 78:  “impairments….leads”

Author Response

Reviewer 1:

Comments and Suggestions for Authors

The stated goal of this manuscript is to provide a systematic review of 17 studies using B12 as a treatment for ASD; the authors also attempt 2 meta-analyses, one focused on biochemical changes subsequent to treatment and the other to an analysis of adverse effects.

Autism spectrum is at present identified entirely clinically and widely understood to be a collection of disorders, rather than a unified entity.  There is anecdotal evidence for the efficacy of B12, with or without other related vitamin cofactors, in the treatment of individual cases.  Accordingly, this is an important topic to review.

Introduction:  could be more accessible to those not already well informed with respect to the topic.  Also, in my opinion, if the main goal of this paper is to review studies using B12 as a treatment, the findings of these studies should be presented in the main body of the paper, not in the introduction:  perhaps ¶ 5 and 6 of introduction should be moved to become 3.1 Summary of studies reviewed.

-- Paragraphs 5 and 6 have been shortened for clarity, and material from paragraph 5 was moved to Section 4.3.

A strength of the manuscript is the robust approach to conducting a systematic study and good enunciation of the steps that were taken to achieve this.

-- We appreciate the reviewer’s comments on the importance of the paper and the strength of the study design.

One significant concern with the design of the research relates to the meta-analysis of biochemical changes.  This was conducted on 3 studies, one with administration of B12 alone and two of B12 + folinic acid with or without betaine.   The extent to which it is appropriate to group these studies for an evaluation of biochemical outcome is not discussed or defended.

-- We have added rationale for combining these 3 studies in section 3.2.2.

Throughout the text, many figures and tables precede, rather than follow, the first mention of same.  This significantly interrupts the flow of reading the communication.  Table 4 is not mentioned in the text at all.

-- We have added a reference to Table 4 in the text. We have also moved each table to the location after the references are discussed in the paper.

Overall, I find this manuscript to be lengthy and rather poorly organized.  Importantly, there is not description of the baseline phenotypes of the patients who did or did not respond to the treatment.  It may well be that this information was not available in the reviewed publications, but it is probably the single most important issue to explore with respect to evaluating the utility of this type of treatment.

-- We have reorganized the paper for better flow and readability. We have added Table 1 which lists the phenotypes for all 17 studies.

Specific issues:

Line 128:  Figure 2 does not list publications, but rather is the flowchart, as mentioned in line 132.  Table 5?  Is this a typo?

-- Table 5 is found in the discussion section, but for clarity we have removed the Table 5 reference from line 128. For clarity, we also removed the reference to Figure 2 in section 2.1 since it is described in section 2.2.

Table 1 is labelled Biochemical changes…  but not all of the outcomes are biochemical in nature.  Table 1 refers to “methylation status” but the text refers to “methylation compacity [sic}.”  Methylation status will mean something different to molecular geneticists, so suggest that methylation capacity might be better.

-- Table 1 has been updated to include only biochemical changes; “methylation status” has been removed from Table 1. We also ensured that there are no other references to “methylation status” in the paper.

Lines 231-235:  Patients with TCN2 mutations may have autistic symptoms, but they also have a genetically defined disorder, and should probably not be grouped with ASD patients not otherwise defined.

-- Since autism is a behaviorally defined disorder independent of genetic abnormalities, we feel it is appropriate to include this ASD case. In fact, the recent definition in the DSM-5 specifies specifically whether the child diagnosed with ASD has a genetic disorder or not.

Discussion:  Lines 524-542:  not clear what the role of 2 additional studies might be here.  If the authors mean to discuss this as a possible mechanism underlying the effects of B12/cobalamin, they need to formulate this part of the discussion differently.

-- These 2 additional studies were moved to the next section “4.3 Biological Mechanisms of Actions.”

Review text for grammatical errors:  eg line 11 (2nd sentence of abstract): no verb; line 78:  “impairments….leads”

-- These grammatical errors have been fixed and we have reviewed the remainder of the text for errors. We appreciate the reviewer’s attention to detail and helpful feedback—we feel this has improved the paper significantly.

Reviewer 2 Report

Every factor related to the treatment of autism and the improvement of the quality of life of patients (here: vitamin B12) should be investigated and made public. Tables (1-5) are of tremendous value, which is a great compilation of information on how vitamin B12 has been used in previous studies. I am also glad that the authors found a description based on double-blind, placebo controlled (DBPC) studies. The manuscript is of high quality and clearly written. I am glad with the amount and reliability of the facts about limitation of published studies. I believe that researchers, parents, physicians and healthcare professionals should be encouraged to read this paper.

Below I present my comments:

1) The absorption of vitamin B12 in the gut is a calcium-dependent process. With calcium deficiency, the absorption process becomes very limited, which can lead to vitamin B12 deficiency. Have the authors found information on this? Were the questionnaires on the daily diet of the patients completed (f.e. including calcium intake)?

2) Have interactions between vitamin B12 and other vitamins been reported? I mean, for example, vitamin D, which is also considered as a factor improving the quality of life of people with autism.

Author Response

Reviewer 2.

Comments and Suggestions for Authors

Every factor related to the treatment of autism and the improvement of the quality of life of patients (here: vitamin B12) should be investigated and made public. Tables (1-5) are of tremendous value, which is a great compilation of information on how vitamin B12 has been used in previous studies. I am also glad that the authors found a description based on double-blind, placebo controlled (DBPC) studies. The manuscript is of high quality and clearly written. I am glad with the amount and reliability of the facts about limitation of published studies. I believe that researchers, parents, physicians and healthcare professionals should be encouraged to read this paper.

-- We appreciate the reviewer’s comment on the quality and clarity of the paper.

Below I present my comments:

1) The absorption of vitamin B12 in the gut is a calcium-dependent process. With calcium deficiency, the absorption process becomes very limited, which can lead to vitamin B12 deficiency. Have the authors found information on this? Were the questionnaires on the daily diet of the patients completed (f.e. including calcium intake)?

-- As far as we can tell none of the studies addressed or examined this finding. However, we added text to section 4.3 concerning this fact.

2) Have interactions between vitamin B12 and other vitamins been reported? I mean, for example, vitamin D, which is also considered as a factor improving the quality of life of people with autism.

-- We did not find any studies that discussed this, but we added a discussion of this to section 4.3. We appreciated the reviewer’s comments which we feel has improved the quality of the paper.

Round 2

Reviewer 1 Report

This manuscript is much improved and I would agree that it is ready for publication